

# Optimization of oil industry wastewater treatment system and proposing empirical correlations for chemical oxygen demand removal using electrocoagulation and predicting the system's performance by artificial neural network

Atef El Jery[1], Hayder Mahmood Salman[2], Nadhir Al-Ansari[3],
Saad Sh. Sammen[4], Mohammed Abdul Jaleel Maktoof[2] and
Hussein A. Z. AL-bonsrulah[5,6]

[1] Department of Chemical Engineering, College of Engineering, King Khalid University, Abha, King Saudi Arabia
[2] Department of Computer Science, Al-Turath University College Al Mansour, Baghdad, Iraq
[3] Civil, Environmental and Natural Resources Engineering, Lulea University of Technology, Lulea, Sweden
[4] Department of Civil Engineering, College of Engineering, University of Diyala, Diyala Governorate, Iraq
[5] Mechanical Power Technical Engineering Department, Al-Amarah University College, Maysan, Iraq., Maysan, Iraq
[6] Department of Computer Techniques Engineering Al Safwa University College, Karbala, Iraq

Corresponding author
Nadhir Al-Ansari,
nadhir.alansari@ltu.se

## ABSTRACT

The alarming pace of environmental degradation necessitates the treatment of wastewater from the oil industry in order to ensure the long-term sustainability of human civilization. Electrocoagulation has emerged as a promising method for optimizing the removal of chemical oxygen demand (COD) from wastewater obtained from oil refineries. Therefore, in this study, electrocoagulation was experimentally investigated, and a single-factorial approach was employed to identify the optimal conditions, taking into account various parameters such as current density, pH, COD concentration, electrode surface area, and NaCl concentration. The experimental findings revealed that the most favorable conditions for COD removal were determined to be 24 mA/cm$^2$ for current density, pH 8, a COD concentration of 500 mg/l, an electrode surface area of 25.26 cm$^2$, and a NaCl concentration of 0.5 g/l. Correlation equations were proposed to describe the relationship between COD removal and the aforementioned parameters, and double-factorial models were examined to analyze the impact of COD removal over time. The most favorable outcomes were observed after a reaction time of 20 min. Furthermore, an artificial neural network model was developed based on the experimental data to predict COD removal from wastewater generated by the oil industry. The model exhibited a mean absolute error (MAE) of 1.12% and a coefficient of determination (R$^2$) of 0.99, indicating its high accuracy. These findings suggest that machine learning-based models have the potential to effectively predict

COD removal and may even serve as viable alternatives to traditional experimental and numerical techniques.

# INTRODUCTION

Petroleum wastewaters and oil-water emulsions are significant contributors to water contamination due to the presence of organic compounds. Various industries, including metalworking, transportation, oil and gas, petrochemical, and refinery, generate wastewater containing petroleum compounds (*Yavuz, Koparal & Öğütveren, 2010*; *Ma et al., 2021*; *Wan et al., 2023*). Refinery and petrochemical industries, in particular, utilize crude oil as a raw material to produce fuels, lubricating oils, and other useful products (*Qu et al., 2023*; *Zhang et al., 2023a*). The refining of crude oil requires a substantial amount of water, and the characteristics of the resulting wastewater depend on the specific refinery processes employed. The wastewater typically contains dissolved or suspended petroleum compounds, including a mixture of hydrocarbons, polyaromatic hydrocarbons, dissolved mineral compounds, chemical compounds, and additives such as anticorrosive, pesticides, emulsion breakers, and antifoams. It may also contain solids like corrosion products, bacteria, wax, and dissolved gases (*Santos et al., 2006*). Aromatic hydrocarbons are a major contributor to the levels of chemical oxygen demand (COD) and ammonia nitrogen in oil effluents (*Yan et al., 2014*; *Wang et al., 2022c*; *Bhagawan et al., 2016*). Different methods of treating wastewater contaminated with petroleum substances are divided into physical, chemical, and biological groups. Popular techniques investigated for the treatment of petroleum wastewater are absorption (*El-Naas, Al-Zuhair & Alhaija, 2010*; *Wang et al., 2021a*), chemical precipitation (*Altaş & Büyükgüngör, 2008*; *Wang et al., 2022a*), and wet oxidation (*Sun, Zhang & Quan, 2008*; *Tian et al., 2022*). Additionally, coagulation and flocculation (*Verma, Prasad & Mishra, 2010*; *Zhang et al., 2023b*), photocatalytic oxidation (*Shahrezaei et al., 2012*), and photon (*Aziz & Daud, 2012*) were utilized recently. He mentioned catalytic vacuum distillation (*Yan et al., 2010*), sequential batch reactor (*Pajoumshariati, Zare & Bonakdarpour, 2017*; *Lv et al., 2021*), and membrane bioreactor (*Razavi & Miri, 2015*; *Chen et al., 2023*). But physical methods are generally inefficient due to time-consuming pollutant transfer to another phase (*Guo et al., 2023*; *Liu et al., 2018*). Biological processes are inefficient due to their low ability to destroy resistant and slowly decomposing petroleum hydrocarbon pollutants. Various treatment methods for petroleum-contaminated wastewater can be categorized into physical, chemical, and biological processes. Commonly investigated techniques include absorption, chemical precipitation, wet oxidation, coagulation and flocculation, photocatalytic oxidation, and membrane-based processes. However, physical methods are often inefficient due to the slow transfer of pollutants to another phase, while biological processes have limited effectiveness in degrading resistant and slowly decomposing petroleum hydrocarbon

pollutants. Electrochemical methods have gained significant attention due to their numerous advantages (*Wang et al., 2023*; *Wang et al., 2022b*; *Shangguan et al., 2022*). Electrocoagulation (EC), also known as electric coagulation, combines the principles of electrochemistry to treat wastewater. By passing an electric current through the fluid, pollutants become destabilized. This electrochemical process generates coagulating agents (such as iron and aluminum hydroxides) that neutralize the electric charge of pollutants and facilitate their removal (*García-García et al., 2015*). The chemical reactions occurring in an electrocoagulation cell can be described by Eqs. (1) to (5) (*Rincón & La Motta, 2014*):

$$\text{In iron anode} \quad Fe \rightarrow Fe^{2+} + 2e \tag{1}$$

$$\text{In alkaline} \quad Fe^{2+} + 3OH^{-} \rightarrow Fe(OH)_2 \tag{2}$$

$$\text{In acid} \quad 4Fe^{2+} + O_2 + 2H_2O \rightarrow 4Fe^{3+} + 4OH^{-} \tag{3}$$

$$\text{Oxygen production reaction} \quad 4e + 2H_2O \rightarrow O_2 + 4H^{+} \tag{4}$$

$$\text{Reaction at the cathode} \quad 2H_2O + 2e \rightarrow H_2 + 2OH^{-} \tag{5}$$

$Fe^{2+}$ or $Fe^{3+}$ ions combine with water and hydroxyl ions and form various hydroxides and polyhydroxides. In the case of iron, $Fe(OH)_2$, $Fe(OH)_3$, $Fe(OH)_4^{-}$, $Fe(OH)_2^{+}$, $Fe(OH)^{2+}$, and $FeO(OH)$ are formed. The production of metal ions in the anode and hydrogen gas is done in the cathode. Metal ions form clots keep these particles floating by trapping pollutants and hydrogen gas (*Linares-Hernández et al., 2010*; *Ji et al., 2023*). This approach reduces pollution load by eliminating suspended solids and depositing soluble organic molecules into organometallic compounds (*Murugananthan, Raju & Prabhakar, 2004*; *Wang et al., 2021b*). This method is proven efficient in purifying petroleum compounds, so the following studies are presented.

*El-Naas et al. (2009)* treated two samples of petroleum wastewater with an initial COD concentration of 599 and 4,050 mg/L and sulfate levels of 887 and 1,222 mg/L, respectively, using the electrocoagulation method. For the first sample, the COD removal efficiency was 93%, and the sulfate was 93%; for the second sample, the COD removal efficiency was 42%, and the sulfate was 24%. The optimal test conditions were pH equal to 8, and the intensity of the current density was 13 $mA/cm^2$. *El-Ashtoukhy et al. (2013)* used an electrocoagulation system with a fixed bed reactor to eliminate phenolic compounds in the refinery wastewater. They also mentioned the best case for removing pollutants in their study. *Hariz et al. (2013)* achieved a reduction of more than 80% in sulfide wastewater treatment utilizing six electrodes with dimensions of 5 ×10 cm in flow density of 21 $mA/cm^2$, the pH of 9, 129 mS/cm electrical conductivity in 30 min, and a COD concentration of 72,450 mg/L. *Elnenay et al. (2017)* used electrocoagulation to remove organic petroleum compounds from wastewater, such as drilling fluids. Electrochemical cell with dimensions of $12 \times 12 \times 15$ $cm$, in which the cathode electrode is made of stainless steel with dimensions of $9 \times 9$ $cm$ and anode electrodes are aluminum

mesh-framed with dimensions of $9 \times 9$ *cm* and an effective surface of 115.2 *cm²* were installed horizontally. The distance between the cathode and anode was considered to be 2.5 cm. The synthetic effluent mixture of 70% water, 30% diesel, and 3% of diesel volume was an emulsifier (Tween 80). A study (*Bozorgnezhad et al., 2015*) was conducted on a single-serpentine transparent PEMFC to examine water management in the cathode channel at different stoichiometry, RH, and temperature. Water coverage ratio was used to measure liquid water accumulation, which was found to be significant in the elbows and later channel rows near the gas outlet. *Bozorgnezhad et al. (2015)* analyzed water management in the cathode channel of a single-serpentine transparent PEMFC under varying conditions of stoichiometry, relative humidity, and temperature. The researchers used water coverage ratio to measure the accumulation of liquid water, which was found to be concentrated in the elbows and later rows near the gas outlet. The study concluded that cathode stoichiometry had a greater impact on water management and cell performance than anode stoichiometry. Additionally, the study examined the effect of water coverage on cell performance and determined the time durations for different two-phase flow patterns.

*Adjeroud-Abdellatif et al. (2022)* demonstrated that ultrasound-assisted extraction (UAE) is a more efficient technique for extracting opuntia ficus-indica (OFI) cladode mucilage compared to conventional extraction (CE). The incorporation of OFI mucilage into the EC-EF water treatment process resulted in enhanced turbidity removal efficiency. The mucilage was identified as a polysaccharide-based biomaterial with functional groups, and its inclusion significantly improved the efficiency of the EC-EF process. *Syam Babu et al. (2020)* provided evidence that electrocoagulation (EC) is a highly effective method for removing chemical oxygen demand (COD) and color from industrial wastewater, achieving removal efficiencies exceeding 80% for most types of wastewater. Furthermore, the EC process requires less energy than other removal methods, making it a cost-effective solution for wastewater treatment, particularly in energy-intensive industries. The study also suggested various methods for the safe disposal of sludge generated by the EC process, ensuring minimal environmental impact. *Valero et al. (2011)* aimed to optimize the electrocoagulation process for treating wastewater from the almond industry. This involved conducting laboratory-scale experiments to analyze the effects of different wastewater characteristics and process variables on removal efficiencies. After determining the optimal conditions, the researchers scaled up the process to a larger scale to assess its effectiveness. Additionally, the study considered economic parameters to evaluate the cost-effectiveness and practicality of the electrocoagulation process. *Yu et al. (2023)* discovered that utilizing a novel centrifugal electrode reactor in the electrocoagulation (EC) process significantly enhanced the removal efficiency of heavy metals compared to stationary electrodes. This indicates the crucial role of centrifugal electrodes in improving the treatment of heavy metal wastewater. Electrochemical analysis revealed that the anodic polarization behavior of the aluminum anode in the centrifugal electrodes exhibited dissolution characteristics instead of passivation, thanks to the enhanced diffusion of Cl– ions, which reduced anode passivation. Additionally, kinetics analysis demonstrated that the removal of heavy metals in the EC process using centrifugal electrodes followed the Variable-Order-Kinetic (VOK) model based on Langmuir adsorption, indicating the

variable-order reaction of heavy metal adsorption onto the electrode surface. *Yang et al. (2022)* demonstrated that fe-electrocoagulation (Fe-EC) and Al-electrocoagulation (Al-EC) processes effectively removed phosphate from real domestic wastewater, achieving a removal efficiency of 98% ± 2% for Fe-EC under both low and high dissolved oxygen concentrations. The composition of the flocs varied, with Fe-EC under low dissolved oxygen (DO) forming green rust, Fe-EC under high DO forming amorphous trivalent iron oxide/hydroxide, and Al-EC forming amorphous alum hydroxide. The removal mechanisms differed, with coagulation being the primary mechanism for Fe-EC under high DO and Al-EC, while ion-exchange adsorption played a significant role in Fe-EC under low DO. *Xu et al. (2022)* focused on utilizing an interpenetrating bipolar plate electrocoagulation (IBPE) reactor to remove microplastics and heavy metals from wastewater, addressing the common issue of multiple pollutants. The IBPE reactor exhibited high removal efficiency, achieving rates of 95.16% for heavy metals and 97.5% for microplastics, demonstrating its effectiveness. The study optimized process parameters such as current density, initial pH, and reaction time to achieve the highest removal efficiency, while mathematical modeling and analysis of complexation forms and surface groups provided insights into the removal mechanisms. Furthermore, cost analysis highlighted the potential feasibility and cost-effectiveness of the IBPE reactor for large-scale applications in reducing the environmental impact of pollutants.

The use of artificial intelligence in electrocoagulation is overlooked. There is little research using machine learning algorithms to predict a parameter in wastewater treatment, but this trend has changed in the last few years (*Yaqub & Lee, 2022*; *Zaboli, Alimoradi & Shams, 2022*; *Alimoradi & Shams, 2019*; *Alimoradi, Shams & Ashgriz, 2023*, *2022*). The use of artificial intelligence is becoming more and more common in engineering problems due to its simplicity of application and accuracy. *Zhu et al. (2021)* utilized long-term and short-term memory to predict the removal of electrocoagulation. *Miao et al. (2021)* studied an intelligent sewage treatment. They used machine learning algorithms such as SVM, LSTM, and GRU to predict the plant's outflow. According to their results, the GRU model is the best. *Yaqub et al. (2020)* used a long short-term memory neural network to predict the removal efficiency of a plant's wastewater. They showed that the proposed model had promising results.

Despite the increasing interest in electrocoagulation, there remains a literature gap concerning the optimization of this method specifically for the treatment of oil industry wastewater. This study aims to address this gap by experimentally investigating the use of electrocoagulation and identifying the optimal conditions for COD removal.
By considering parameters such as current density, pH, COD concentration, electrode surface area, and NaCl concentration, the study employs a single-factorial approach to identify the most favorable conditions. In addition to the optimization aspect, the study explores the impact of COD removal over time using double-factorial models.
By analyzing the outcomes at different reaction times, the research provides valuable information on the time-dependent behavior of COD removal using electrocoagulation. This research goes beyond traditional experimental and numerical techniques by
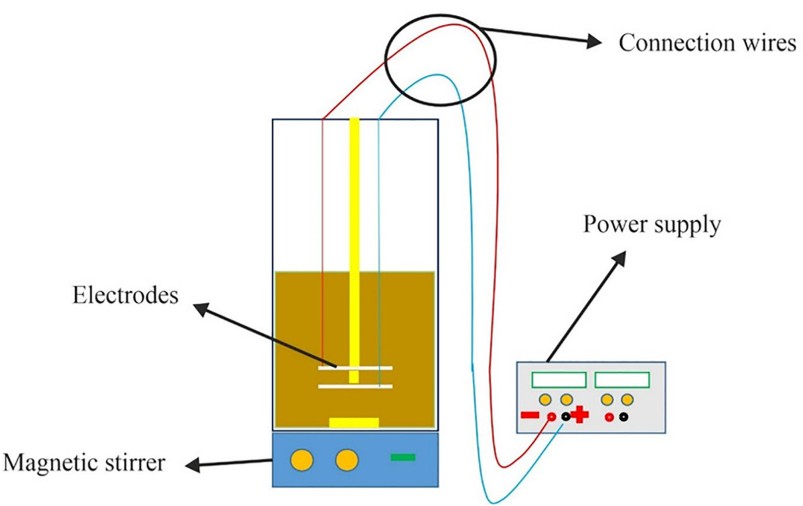

**Figure 1 The schematic model of the experiment.**

developing an artificial neural network (ANN) model to predict COD removal from oil industry wastewater.

According to the investigations, in most research in the electrocoagulation process, the arrangement of the electrodes is vertical. In this research, they are installed horizontally to use the flotation power, and the optimization is by the single factorial method. Also, the experiment design was considered with three essential parameters: removal efficiency, electrode dissolution, and energy consumption. In the present work, we aim to investigate the electrochemical process to remove petroleum hydrocarbons effectively to improve the environmental situation. Investigating the effect of the electrode surface, placing the electrodes horizontally, and investigating the impact of flotation on the process used to treat petroleum wastewater are among the innovations of this research. Also, a double-factorial optimization method is utilized to determine the best range of initial COD concentration, electrode surface area, pH, current density, NaCl concentration, and time for the optimum COD removal. Additionally, a machine learning algorithm is employed to propose a predictive model for COD removal. This model incorporates a hyperparameter tuning procedure, with which the model could be optimized to predict the results as accurately as possible.

## MATERIALS AND METHODS

Based on Fig. 1, an electrochemical cell made of Plexiglass was used for experiments involving discontinuous current. The cell had dimensions of $16 \times 16 \times 25$ cm and an effective volume of four liters. The cathode and anode were horizontal electrodes made of SAE 304 stainless steel with 99% purity, and they were positioned 1 cm apart with four effective surfaces. The anode was placed 6 cm away from the container's surface. One important factor in particle suspension within the solution is hydrogen production, which required the current to be connected in a way that positioned the cathode electrode at the top and the anode at the bottom.

Stainless steel electrodes are preferred over aluminum and steel electrodes in electrocoagulation due to their higher resistance to corrosion and longer lifespan. Aluminum electrodes can corrode rapidly in the presence of chloride ions commonly found in wastewater. Steel electrodes can also corrode and produce iron ions that can interfere with the coagulation process. On the other hand, stainless steel is highly resistant to corrosion and does not generate unwanted ions, making it a more reliable choice for electrocoagulation.

For the experiments, the electrodes were cleaned using distilled water and a weak acid after each use. A DC power supply was used to provide the electric current, and a magnetic stirrer was employed to ensure homogeneity of the solution. It is important to note that the stirring process was stopped 2 min before taking the sample to prevent any clots formed in the solution from entering the sample. Additionally, a residence time of 30 min was considered for settling to account for any small amount of clots that may have entered the sample and to minimize resulting errors. All experiments were conducted at room temperature, and they were repeated three times to ensure the reliability of the data.

Energy consumption is a significant factor in these processes due to electricity usage. Therefore, apart from examining the removal of chemical oxygen demand (COD) in this process, two parameters, electrode dissolution and energy consumption, were evaluated to determine the optimal factors affecting the process. The specific energy consumption, measured in kilowatt-hours per kilogram of COD removed, is an important metric for justifying the use of this process. It was calculated using Eq. (6), where U represents voltage, I represents current intensity, $C_0$ and C represent the concentration at the start of the reaction and a specific time after the start of the test, V represents the volume of wastewater, and t represents the reaction time (*Demirbas & Kobya, 2017*).

$$SEC = (U \times I \times t)/V \times (C_0 - C) \tag{6}$$

The amount of dissolution and, as a result, decomposition of the anode metal depends on the reaction time and the amount of electric current going through the effluent. This parameter is measured using Faraday's law according to Eq. (7). In this equation, m is the mass of the dissolved metal (g), t is the duration of electrolysis (s), M is 55.84 g/mol, F is is 96485 $C/mol$, and Z is equal to 2 (*Hu et al., 2016*).

$$m = \frac{I \times t \times M}{F \times Z} \tag{7}$$

Most of the petroleum compounds in the wastewater of oil refineries are in the range of the magnetic stirrer to mix and homogenize (*dos Santos et al., 2017*). In order to carry out the current study, diesel and crude oil were used to synthesize wastewater similar to refinery wastewater with a ratio of 1 to 2. Also, a cationic surfactant named cetyltrimethylammonium bromide was used to prepare synthetic wastewater for properly mixing oil and diesel in water. A certain amount of pollutant (depending on the desired concentration) and 0.05 grams of surfactant were vigorously stirred in water for 10 min to prepare synthetic wastewater. Then the resulting mixture was kept in a decanter for 5 min to separate oily and undissolved oil compounds from the solution entering the systems.

**Table 1 The properties of the wastewater.**

| Parameter | Range |
| --- | --- |
| pH (−) | 4–10 |
| Current density ($mA/cm^2$) | 3–35 |
| NaCl concentration ($g/L$) | 0.3–2 |
| Initial COD concentration ($mg/L$) | 200–2,400 |
| The surface area on the electrodes ($cm^2$) | 25.26–79.26 |

An additional amount of pollution was added to the solution due to the existence of the surfactant in different concentrations of COD. This amount equals 50 mg/L.

The range of parameters examined in this process is according to similar research (*El-Naas et al., 2009*; *Elnenay et al., 2017*; *Chen, 2004*; *Abdelwahab, Amin & El-Ashtoukhy, 2009*; *Alimoradi, Shams & Valizadeh, 2017*; *An et al., 2017*; *Moussa et al., 2017*), and the quality of primary wastewater is presented in Table 1.

In this study, synthetic wastewater was prepared using crude oil and diesel. To provide electrical conductivity, a cationic surfactant called cetyltrimethylammonium bromide (Sigma-Aldrich, Shanghai, China) and sodium chloride (Merck) were used as a carrier electrolyte. Sodium hydroxide (NaOH) with a concentration of 0.1 normal was used to adjust the pH of the solution. For COD analysis, potassium dichromate, mercury sulfate, silver sulfate, and potassium hydrogen phthalate (Merck) were employed. pH adjustment and COD analysis were carried out using 98% sulfuric acid. To determine the key parameters and conduct tests, various instruments were utilized. The Hach spectrophotometer model DR4000 and Hach COD reactor model DRB200 were used to calculate COD using the Closed Colorimetric method, Reflux (number 5220D), based on the standard book of water and wastewater tests (*American Public Health Association, 1926*). The Martini EC meter model MI 805 and Metrohm pH meter model PJ300 were used for measuring electrical conductivity and pH, respectively. The Sugon power supply model 3005D provided the necessary electric current. The Ika magnetic stirrer model RH-Bassic2 ensured homogeneity of the solution. The Sigma 8-branch centrifuge, 55-liter digital steel oven, and Mettler digital scale model PJ300 were employed for centrifugation, drying, and weighing purposes, respectively.

## RESULTS

The results of the experiments of the electrocoagulation system to optimize the five parameters of the electrode surface, the initial pH concentration of the initial COD, the electric current density, and the electric conductivity of the solution to increase the COD removal efficiency are presented below.

In the tests related to changes in the initial concentration of the input to the system, according to Fig. 2, COD removal efficiency after 12 min from the start of the experiment in initial concentrations of 200, 500, 1,000, 1,500, and 2,400 $mg/L$ is equal to 73.71%, 69.75%, 71.18%, 9.17%, and 12.97%, respectively. As can be seen, increasing the concentration from 200 to 2,400 $mg/L$ decreases the removal rate, and more time is

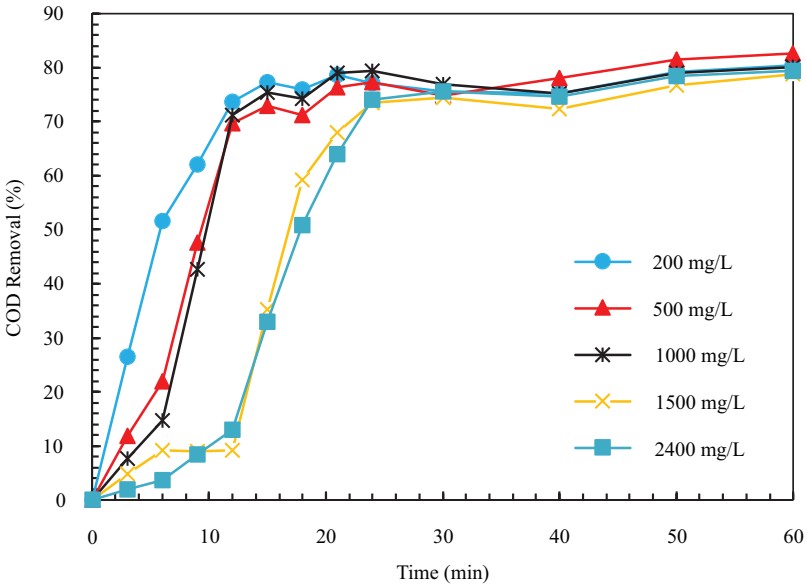

**Figure 2** The COD removal percentage for different initial COD concentration with respect to time (pH = 8, ESA = 25.26 cm$^2$, i = 24 mA/cm$^2$, NaCl concentration = 0.5 g/l).

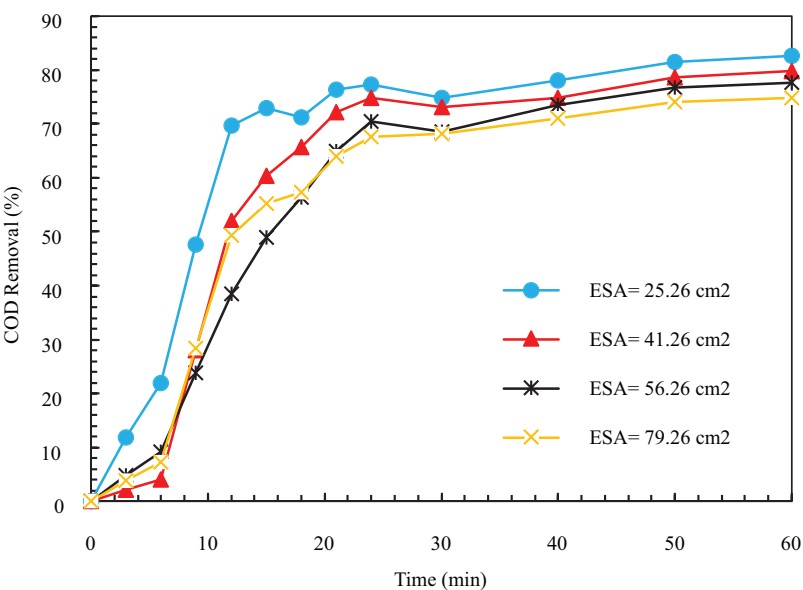

**Figure 3** The COD removal percentage for various ESA with respect to time (pH = 8, NaCl concentration = 0.5 g/l, I = 1 A, Initial COD = 500 mg/l).

needed to reach a constant removal efficiency. The reason for the decline in removal speed with the increase in the amount of pollutant is that for a constant electric current density, the amount of the produced metal hydroxide coagulant is constant with respect to time. As a result, it is not enough to coagulate particles at higher concentrations (*Abdelwahab, Amin & El-Ashtoukhy, 2009*). The initial COD concentration of 500 $mg/L$ was chosen as the optimal case, with 82.63% COD removal in 60 min.

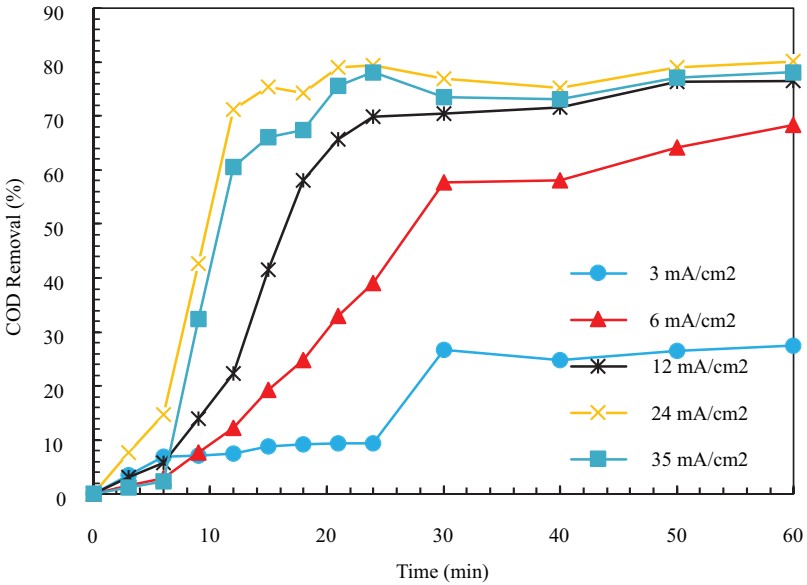

**Figure 4 The COD removal percentage for different current density with respect to time (pH = 8, ESA = 25.26 cm², Initial COD = 1,000 mg/l, NaCl concentration = 0.5 g/l).**

Tests were performed at four different levels, and based on the initial tests, the values of other parameters were considered constant to find the optimal electrode surface area (ESA). As seen in Fig. 3, the removal process's speed decreases with the ESA's increase. The COD removal efficiency for electrode surfaces in 25.26, 41.26, 56.26, and 79.26 cm² after 12 min was equal to 69.75%, 52.02%, 38.42%, and 49.36%. The same results after 1 h were 82.63%, 79.82%, 77.64%, and 74.83%, respectively. The decrease in efficiency and removal with respect to time with the increase of the ESA can be because the produced bubbles are held under the electrode surface. As the production continues and they stick together, larger bubbles are formed, which are released. Additionally, as these bubbles move upward, they become larger; hence, they cannot separate the small clots in the sewage.

In Fig. 4, the COD removal after 12 min for current densities of 3, 6, 12, 24, and 35 $mA/cm^2$ is equal to 7.53, 12.24, 22.29, 71.18, and 6.58%. The COD removal after 60 min was 27.48%, 68.35%, 76.49%, 80.1%, and 78.16%. In Fig. 4, the removal percentage augmented significantly by rising current from 3 to 24 $mA/cm^2$. After that, the COD removal decreased with the augment of this parameter. At 34 mA/cm², after 60 min from the start of the reaction, the removal efficiency reached a constant value of 80.1%.

In general, as in Fig. 4, the COD removal grows with augmenting the amount of electric current up to an optimal value and then decreases slightly. The primary effect of electric current on this process is twofold. First, it alters the number of metal ions in the anode. Then it changes the $H_2$ production on the cathode. The increase in current would lead to the breakdown of anode electrodes. Also, the $Fe(OH)_3$ increases which destabilize colloidal particles. This is favorable since the separation of the pollutant is accelerated. Moreover, as the $H_2$ production increases, and the bubbles' size decreases, their density and upward flux

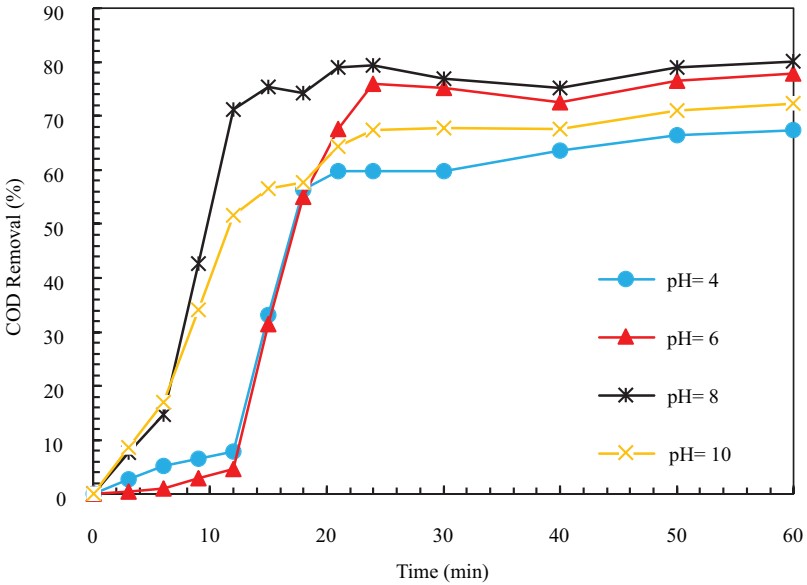

**Figure 5 The COD removal percentage with different pH with respect to time (i = 24 mA/cm², ESA = 25.26 cm², Initial COD = 1,000 mg/l, NaCl concentration = 0.5 g/l).**

augment. As the bubbles become smaller, their contact surface to pollutants increases, and the removal efficiency increases. The produced small particles raise the oil particles in the solution after flocculation and remove them during flotation (*Chen, 2004*; *An et al., 2017*). As can be seen, with the increase in the speed of the process from 24 to 35 mA/cm², the removal is reduced. The increased coagulant clots due to the excessive increase in flow lead to disruption of the proper oxidation reaction for oxygen production. That is why excessive current density values are unsuitable for achieving high removal efficiency (*Hariz et al., 2013*). Finally, 24 mA/cm² (1 A) is the best case due to the removal of 80.1% in 60 min, and the experiments were continued with an electric current of 1 A.

In investigating the effect of different pHs, the COD removal efficiency after 12 min at pHs 4, 6, 8, and 10 was obtained as 7.78%, 4.67%, 71.18%, and 51.71%, respectively.

According to Fig. 5, the speed of the removal process at pH equal to 8 was higher than other values, so after 12 min, it reached a high rate of 71.18%. As can be seen, COD removal efficiency at a pH equal to 8. The pH of wastewater has a significant role the coagulation due to its effect on the electrical conductivity of the solution, electrode breakdown, produced hydroxide species, and the zeta potential. In different acidic, alkaline, and neutral environments, various cations and hydroxides are formed to destabilize the particles; hence, the coagulation depends on the pH of the environment (*Sahu, Mazumdar & Chaudhari, 2014*). It is hydrolyzed as an insoluble iron compound depending on the environment's pH and the cell's potential. The results of different researchers regarding the mechanism of electrochemical dissolution of iron anodes are contradictory, and so far, there are no correct experimental results for the iron species formed during the electrocoagulation process.

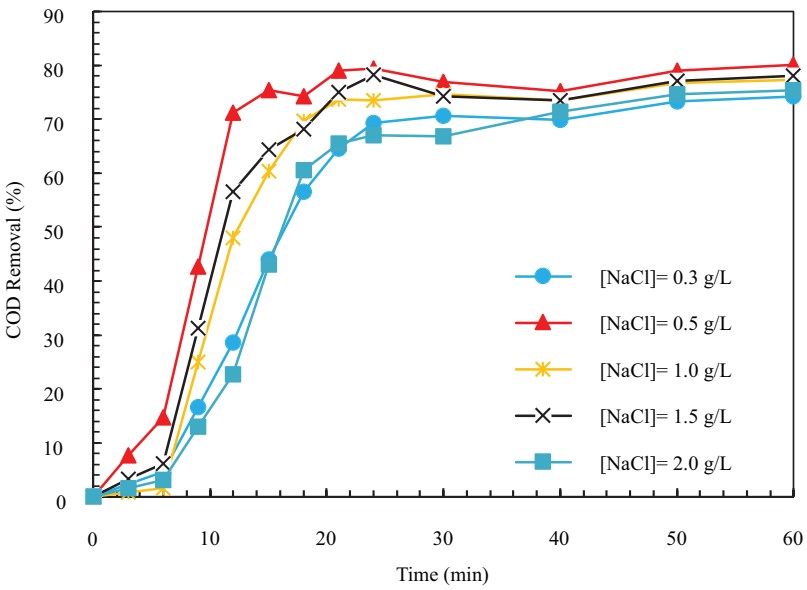

**Figure 6 The COD removal percentage for different NaCl concentrations with respect to time (i = 24 mA/cm², ESA = 25.26 cm², Initial COD = 1,000 mg/l, pH = 8).**

The results of findings of *Lakshmanan, Clifford & Samanta (2009)* indicate that as a result of iron electrolysis, $Fe^{2+}$ is formed, and then this ion is converted into $Fe^{3+}$ due to oxidation with oxygen and suitable pH. Finally, it turns into insoluble monomers $Fe(OH)_{3(S)}/FeOOH_{(s)}$. The oxidation rate of $Fe^{2+}$ becomes slow in low pH, which makes the augmentation of pH of the solution and forming of a mixture of soluble ferrous ions, and $Fe(OH)_{3(S)}/FeOOH_{(s)}$ becomes insoluble. At higher pH (approximately 8.5), The ferrous ion is entirely oxidized, and $Fe(OH)_{3(S)}/FeOOH_{(s)}$ is precipitated. It is proved that the ferrous ions are oxidized to ferric ions at a pH higher than five. Also, it is observed that the complete form of oxidization would happen in pH 8–9. Therefore, the pH suitable for the anode is 5 to 9, and the initial suitable pH to ensure the complete oxidation of ferrous ions, which due to their high solubility, have weak coagulation properties and are incapable of absorbing pollutants, is in the range of 8 to 9. Also, at very high pH, ferrous ions $Fe(OH)_4^-$ are formed, which have weak coagulation properties and pollutant absorption power and reduce the performance of the electrocoagulation process (*Moussa et al., 2017*). *Pérez et al. (2016)* investigated the pH range of 3 to 9 in electrocoagulation. They found the reaction's optimal pH to be the wastewater's natural pH (6.5), with a COD removal efficiency of 88%.

In the process of electrolysis, electrical conductivity is a pretty influential parameter in COD removal and electricity consumption, and the operating cost is directly related to it. In the electrocoagulation process, to establish an electric current, the solution needs minimum conductivity, which is realized by adding salts such as sodium sulfate and sodium chloride (*Khandegar & Saroha, 2013*). As the sodium chloride concentration augments, the electrical conductivity of the solution increases. To determine the optimal value of the electrical conductivity parameter, we conducted experiments at 0.3, 0.5, 1, 1.5,

**Table 2 The constant values in Eq. 11 for different y and the evaluation of the correlations using RSME and R^2.**

| Values | $A_1$ | $A_2$ | $A_3$ | $A_4$ | $A_5$ | $A_6$ | $A_7$ | $A_8$ | $A_9$ | $A_{10}$ | RMSE | $R^2$ |
|---|---|---|---|---|---|---|---|---|---|---|---|---|
| Initial COD concentration | 10.83 | 5.758 | 0.00274 | −0.1441 | 0.00044 | −2.677e−05 | 0.00106 | 3.165e−06 | −1.207e−07 | 8.326e−09 | 10.6 | 0.89 |
| ESA | 27.69 | 5.937 | −1.372 | −0.1502 | 0.01612 | 0.01057 | 0.001093 | 0.000144 | −0.0002213 | 1.123e−05 | 6.86 | 0.95 |
| Current density | −24.43 | 2.165 | 3.609 | −0.05612 | 0.1688 | −0.1547 | 0.0005337 | −0.001619 | −0.001668 | 0.001826 | 10.57 | 0.89 |
| pH | 182 | 4.033 | −102.6 | −0.09454 | 0.3423 | 16.36 | 0.0006671 | −0.00138 | −0.02245 | −0.7945 | 11.14 | 0.88 |
| NaCl concentration | −29.69 | 6.31 | 64.52 | −0.1399 | −0.3182 | −50.23 | 0.0009943 | 0.001989 | 0.1164 | 10.38 | 9.813 | 0.90 |

and 2 g/L of sodium chloride salt concentration with electrical conductivity values of 650, 1,100, 2,200, 3,300, and 4,400 $\mu S/cm$, respectively. As seen in Fig. 6, the COD removal increases with the augmentation of dissolved sodium chloride concentration. After a certain value, the removal drops with the rise in the value of this parameter. In other words, for the concentration of sodium chloride salt 0.3, 0.5, 1, 1.5, and 2 g/L within 12 min, the removal efficiency is 28.64, 71.18, 48.06, 56.49, and 22.78%, respectively. After 1 h, the removal efficiencies for the mentioned sodium chloride were 74.23, 80.1, 77.32, 78.09%, and 75.38%, respectively.

The reason for increasing the pollutant removal speed with increasing salt concentration is that with increasing salt concentration and chloride ion concentration, these ions act as oxidizing agents and remove the passive oxide layer on the anode, which limits its dissolution. Hence, using higher amounts of the metal hydroxide ions improves the removal efficiency. Chloride ion $Cl^-$ in the electrode is transformed into hypochlorite $ClO^-$ by consuming oxygen (*Elnenay et al., 2017*). In other words, $Cl^-$ ion leads to increased anode dissolution decomposition in oxidation and dissolution. This ion produces $Cl_2$ that dissolves in the solution and becomes $ClO^-$. These reactions are according to relations 8 to 10 (*Hanafi, Assobhei & Mountadar, 2010*).

$$2Cl^- \rightarrow Cl_2 + 2e \tag{8}$$

$$Cl_2 + H_2O \rightarrow HClO + H^+ + Cl^- \tag{9}$$

$$HClO \rightarrow ClO^- + H^+ \tag{10}$$

According to Fig. 6, with the increase of salt concentration from 0.3 to 2 g/L, the COD removal drops from 80.1 to 74.9%, and the pollutant removal speed decreases. At high concentrations of NaCl, the excessive dissolution of iron leads to an unfavorable effect from the contact of the coagulant and particles (*An et al., 2017*). In fact, by using higher sodium chloride, $Cl^-$ ions in the solution react with $Fe(OH)_3$ and produce $Fe(OH)_2Cl$, $Fe(OH)Cl_2$ and $FeCl_3$. The new products finally combine with other $Cl$ ions in the form of $FeCl_4^-$. Therefore, the amount of coagulant and $Fe(OH)_3$ and the COD removal decreases (*Khandegar & Saroha, 2013*). When using NaCl salt as an electrolyte, wastewater is treated through coagulant formation and decomposition (*Alimoradi, Shams & Valizadeh, 2017*).

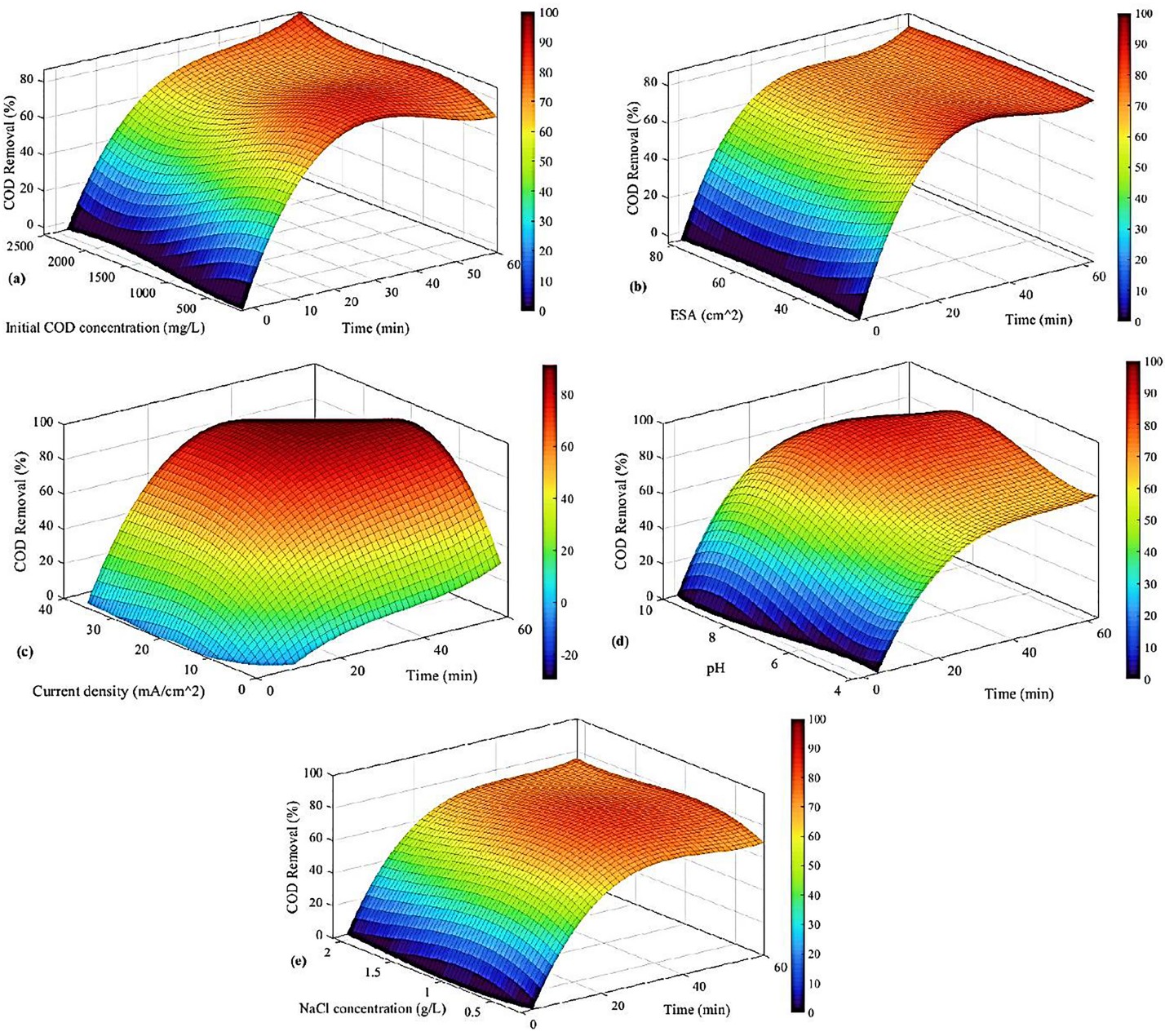

**Figure 7 COD removal based on (A) initial COD concentration and time, (B) ESA and time, (C) current density and time, (D) pH and time, and (E) NaCl concentration and time.**

On the other hand, the presence of chloride ions with sodium chloride electrolyte is also effective in water disinfection (*Moussa et al., 2017*). *Elnenay et al. (2017)* used an electrochemical cell with dimensions of $12 \times 12 \times 15$ cm to treat wastewater containing 70% of water, 30% of diesel, and 3% of diesel volume emulsifier (Tween 80). In their study, the cathode electrode was made of stainless steel with dimensions of $9 \times 9$ cm (*Chen, 2004*). *Hariz et al. (2013)* treated sulfide wastewater from an oil refinery and achieved a COD removal rate of more than 80 percent reduction was achieved.

**Table 3 The optimum case with the best results.**

| Parameters | ESA ($cm^2$) | Initial COD concentration ($mg/L$) | pH ($-$) | Current density ($mA/cm^2$) | NaCl concentration ($g/L$) |
|---|---|---|---|---|---|
| Values | 25.26 | 500 | 8 | 24 | 0.5 |

In this section, high-accuracy empirical correlations for COD removal are introduced. These models are based on time and four other effective parameters. The general polynomial equation of the models is shown as Eq. (11).

$$COD\,removal = A_1 + A_2x + A_3y + A_4x^2 + A_5xy + A_6y^2 + A_7x^3 + A_8x^2y + A_9xy^2 + A_{10}y^3 \quad (11)$$

In Eq. (11), x is time, and y is the other parameters studied in the present study. The values for the constants for each y are presented in Table 2.

The optimized values of COD removal are presented in Fig. 7 based on the correlation of Eq. (11). In this analysis, time is the second factor, so the highest value is not necessarily the best, and we should also consider time.

In Fig. 7A, the highest COD removal is observed when the initial COD concentration is higher than 1,800 mg/L and when the time is equal to 60 min. However, the sharp increase from the beginning of the process until 20 min is mainly the greatest improvement; from then on, COD removal has increased by only 4%. However, the time is almost three times. Therefore, the best case is around 20 to 30 min and an initial concentration of 500 to 800 mg/L. Generally, the COD removal increased as time passed, and a similar trend was exhibited with the rise in initial COD concentration.

In Fig. 7B, the effect of ESA and time is investigated. It is shown that ESA affects COD removal slightly, and time plays a more important role there. This is why the optimum case for COD removal is when time reaches its high values, *i.e.*, 60 min. The effect of ESA is not considerable, and the results are quite the same in all the ESA values. The rate at which COD removal increases is higher in the first 20 min before reaching a local maximum. Then it slightly decreases (about 3%) after 1 h. Therefore, the affordable choice is considering only 20–30 min at relatively low ESAs (smaller than 40 $cm^2$).

The impact of current density and time is studied in Fig. 7C. The result shows that the COD removal augmented as the current augmented. The same is observed for the time, so the optimum case is when the time is higher than 20 min, and the current density is higher than 20 $mA/cm^2$.

Figure 7D indicates the effects of pH and time, where the best case is around the pH of 8 and the time higher than 40 min. It is understood that by increasing the pH, COD removal is augmented. Moreover, as time increased, the same happened.

Finally, in Fig. 7E, the effect of NaCl concentration is studied. The best result for this case is observed when this parameter is between 0.5 to 1 g/L at 60 min. It should be noted that although the optimized case is mentioned, the NaCl concentration does not have very sharp effects on COD removal.

**Table 4  Hyperparameter tuning for layer structure.**

| Model | Layer structure | Mean absolute error (%) | $R^2$ |
|---|---|---|---|
| 1 | (32) | 1.98% | 0.96 |
| 2 | (32,64) | 1.67% | 0.97 |
| 3 | (32,64,32) | 1.61% | 0.97 |
| 4 | (32,64,64,32) | 1.53% | 0.98 |
| 5* | (32,64,128,64,32) | 1.39% | 0.98 |
| 6 | (32,64,128,128,64,32) | 1.48% | 0.98 |
| 7 | (32,64,128,256,128,64,32) | 1.47% | 0.98 |
| 8 | (32,64,128,256,256,128,64,32) | 1.45% | 0.98 |
| 9 | (32,64,128,256,512,256,128,64,32) | 1.49% | 0.98 |

Note:
An asterisk (*) indicated the selected setting.

**Table 5  Hyperparameter tuning for activation function.**

| Model | Activation function | Mean absolute error (%) | $R^2$ |
|---|---|---|---|
| 1 | Linear | 1.87% | 0.96 |
| 2* | ReLU | 1.39% | 0.98 |
| 3 | Sigmoid | 1.46% | 0.98 |

Note:
An asterisk (*) indicated the selected setting.

**Table 6  Hyperparameter tuning for batch size.**

| Model | Batch size | Mean absolute error (%) | $R^2$ |
|---|---|---|---|
| 1 | 2 | 1.98% | 0.96 |
| 2 | 4 | 1.47% | 0.98 |
| 3 | 8 | 1.36% | 0.98 |
| 4 | 16 | 1.39% | 0.98 |
| 5* | 32 | 1.25% | 0.99 |
| 6 | 64 | 1.86% | 0.96 |

Note:
An asterisk (*) indicated the selected setting.

In Table 3, the optimal values resulting from the optimization of the parameters by the single factorial method are presented. In optimal conditions and according to Eq. (6), the specific energy consumption amount equals 7.3 kWh/kg $COD_{Rem}$. According to Eq. (7), the dissolution rate of steel anode within 90 min and 94% COD removal efficiency equals 0.4 kg Fe/kg $COD_{Rem}$.

# DISCUSSION

In order to mitigate the need for further experiments and numerical simulations, machine learning algorithms are utilized to propose predictive models (*Eskandari et al., 2022*; *Chamgordani et al., 2019*; *Sarkar, Biswas & Kundu, 2022*). These algorithms have proven to be very accurate. Among these methods, the artificial neural network (ANN) has stood

**Table 7 Hyperparameter tuning for epochs.**

| Model | Epochs | Mean absolute error (%) | $R^2$ |
|---|---|---|---|
| 1 | 1,500 | 3.21% | 0.89 |
| 2 | 2,500 | 2.76% | 0.92 |
| 3 | 6,000 | 1.96% | 0.96 |
| 4 | 15,000 | 1.55% | 0.98 |
| 5 | 25,000 | 1.38% | 0.98 |
| 6 | 35,000 | 1.19% | 0.99 |
| 7* | 45,000 | 1.12% | 0.99 |
| 8 | 55,000 | 1.29% | 0.98 |

**Note:**
An asterisk (*) indicated the selected setting.

**Table 8 The ANN settings for COD removal.**

| Hyperparameter | Value |
|---|---|
| Layers' structure | (32,64,128,64,32) |
| Batch size | 32 |
| Epochs | 45,000 |
| Activation function | ReLU |
| Learning rate | 0.01 |

out (*Matheri et al., 2021*), so in the present study, we have utilized this algorithm to create models for COD removal. The ANN models are first required to be trained, so the results from the present study are divided into two sections. The first part comprises 70% of the data and is used for training the ANN model. The rest of the data is employed for the evaluation of the performance of the model. In the process of training the model, the are some parameters that require optimization. This process is called hyperparameter tuning and is completely done for the following model. The input parameters considered for this model are NaCl concentration, current density, pH, ESA, initial COD concentration, and time. The process of hyperparameter tuning is done for different parameters such as hidden layers, batch size, epochs, and activation functions. Tables 4–7 present the mentioned comparison study.

As is clear, the chosen model has the lowest MAE and highest $R^2$. Consequently, the selected layer structure is used, and in the next step, presented in Table 5, activation functions are investigated.

Based on the results, the ReLU function seems to be the best choice for the final model due to its low MAE and $R^2$. A similar study is conducted on the batch size, presented in Table 6.

The results show that 32 is the best value for the batch size, so this is selected for the final model. Finally, the number of epochs is investigated to find the best result for this parameter, shown in Table 7.

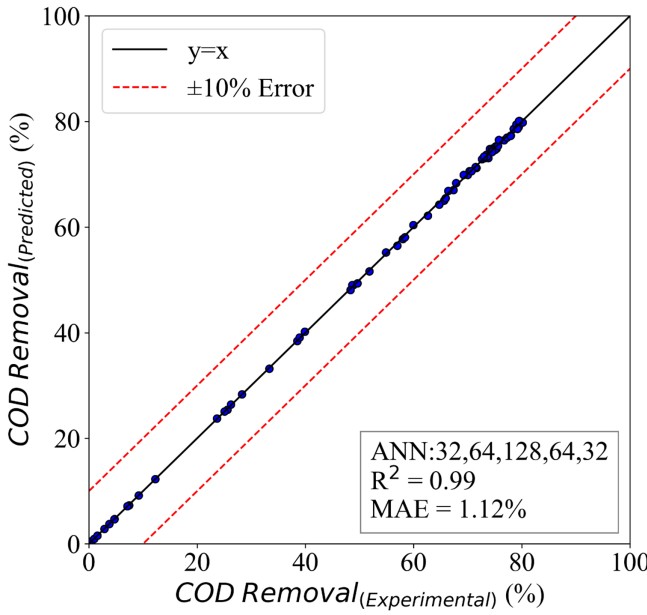

**Figure 8 The ANN model for COD removal.**

The selected parameter for the number of epochs is 45,000 because it has the least error and the highest $R^2$. After careful examination and setting the right hyperparameter, the final model is presented in Table 8.

If the model is trained so deeply on the training data, overfitting could occur, but in the present study, we utilized dropouts and L2 regularizes to avoid this problem. The results of the model are presented in Fig. 8. This figure presents a comparative study of the predicted and experimental results. It is clear that the model has captured the problem's physics and can understand the variations in the input parameters. The evaluation of the model is done with mean absolute error and R_squared (*Chamgordani, 2022*; *Bordbar, Naderi & Alimoradi Chamgordani, 2021*; *El Jery et al., 2023*; *El Jery et al., 2023*). This COD removal model was able to achieve an MAE of 1.12%, and its $R^2$ is 0.99. Therefore, the model is very accurate, so researchers claim that machine learning-based models could 1 day replace conventional research methods (*Naderi et al., 2021*; *El Jery et al., 2023*).

## CONCLUSIONS

The objective of this study was to investigate the application of electrocoagulation for reducing the pollution load of wastewater containing petroleum substances, in order to mitigate environmental impact. This process offers several advantages, including the absence of chemical substances, utilization of simple equipment, generation of a small amount of sludge, and short treatment time. The research focused on analyzing the impact of five key parameters: horizontal electrode surface area, initial COD concentration, electric current density, pH, and NaCl concentration. The single factorial method was employed to examine the individual effects of these parameters and determine the optimal conditions for COD removal. It was observed that pH and current density had a positive influence on COD removal, while an increase in NaCl concentration, ESA, and initial COD

concentration resulted in a decrease in COD removal efficiency. Moreover, the study also determined the optimal conditions for specific energy consumption and the dissolution rate of the steel anode. Thus, electrocoagulation can be considered as a suitable and efficient method for treating wastewater contaminated with petroleum substances. Correlations were established to predict COD removal and the optimum conditions were identified, taking into account the effect of time. Additionally, an artificial neural network was employed to develop predictive models, which demonstrated a high level of accuracy with an $R^2$ value of 0.99.

## ACKNOWLEDGEMENTS

The authors acknowledge the anonymous reviewers for their valuable suggestions that helped improve the quality of the manuscript.

### Funding

This work was financially supported by the Deanship of research of King Khalid University Abha, Saudi Arabia (No. RGP. 2/57/44). The funders had no role in study design, data collection and analysis, decision to publish, or preparation of the manuscript.

### Grant Disclosures

The following grant information was disclosed by the authors:
Deanship of research of King Khalid University Abha, Saudi Arabia: No. RGP. 2/57/44.

### Competing Interests

The authors declare that they have no competing interests.

### Author Contributions

- Atef El Jery conceived and designed the experiments, performed the experiments, analyzed the data, prepared figures and/or tables, authored or reviewed drafts of the article, and approved the final draft.
- Hayder Mahmood Salman conceived and designed the experiments, performed the experiments, analyzed the data, prepared figures and/or tables, authored or reviewed drafts of the article, and approved the final draft.
- Nadhir Al-Ansari performed the experiments, analyzed the data, authored or reviewed drafts of the article, and approved the final draft.
- Saad Sh Sammen performed the experiments, analyzed the data, authored or reviewed drafts of the article, and approved the final draft.
- Mohammed Abdul Jaleel Maktoof conceived and designed the experiments, performed the experiments, prepared figures and/or tables, and approved the final draft.
- Hussein A. Z. AL-bonsrulah performed the experiments, authored or reviewed drafts of the article, and approved the final draft.

## Data Availability

Raw data are available as a Supplemental File.

## Supplemental Information

Supplemental information for this article can be found online at http://dx.doi.org/10.7717/peerj.15852#supplemental-information.

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
