# Peer review of "Optimization of oil industry wastewater treatment system and proposing empirical correlations for chemical oxygen demand removal using electrocoagulation and predicting the system’s performance by artificial neural network"

_PeerJ, doi:10.7717/peerj.15852_

## Round 0.1 · original submission · Major Revisions

Dear authors, please incorporate all comments of reviewers. However, Reviewer 2 has suggested some literature to add. So, please select one or max two papers to cite if these papers are relevant to your study. Otherwise, please ignore all to cite.

Reviewer 1 ·

Basic reporting

1. The writing pattern of the manuscript didn't reach the academic writing level.
2. There are many mistakes in the language of the manuscript. it has to improve.

Experimental design

1. There is no novelty appears in the manuscript.
2, The authors have to justify the reasons for using stainless steel electrodes instead of Aluminum or steel.

Validity of the findings

1. During the discussion of the results it is required to mention the operation condition (current, pH, T,) for each experiment.
2. The result of Fig 2 is not suitable, as it is well known the increase of COD concentration leads to increasing the mass transfer during the operation, which will improve the removal rate. but the result demonstrated the inverse fact I think because the amount of dissolved metals and formation of coagulant is low may be because of the kind of anode material (stainless steel) and the use of low current density.
3. The COD removal during the result description barograph for 500 mg/L initial concentration was fixed in two different values first 69.75 and then 82.63%. The authors have to select which one is correct
4. Describe electrode area effect on the COD removal rate is ambiguous; also it is inversely with electrochemical principles.
5. The discussed effect of current density on COD removal has a sense, but the authors have to justify the decline of the COD removal rate with an increase in the current value to 35 mA/cm^2 .

Reviewer 2 ·

Basic reporting

The paper titled "Optimization of oil industry wastewater treatment system and proposing empirical correlations for COD removal using electrocoagulation and predicting the system’s performance by artificial neural network" is an excellent contribution to the field of environmental engineering. The experimental study with the combination of machine learning is very intriguing. The manuscript possesses the required novelty to be published in a journal.

The authors have done an outstanding job of investigating the performance of an electrocoagulation system for the treatment of wastewater in the oil industry. The paper presents an in-depth analysis of the optimization process of the system and proposes empirical correlations for COD removal. The authors have also used artificial neural networks to predict the performance of the system, which is a novel approach in the field of wastewater treatment.

The paper is well-organized, and the results are presented clearly and concisely. The authors have used appropriate methods to analyze the data, which adds credibility to their findings. The figures and tables included in the paper are also informative and help to support the arguments presented in the text.

However, as a minor revision, I would suggest that the authors include more information on the potential applications of their findings. Specifically, it would be helpful to know how this research could be applied to other industries and wastewater treatment systems. Also, it is needed that authors mention the application of water management and water treatment in other environment friendly systems, such as renewable energies and specifically the PEM fuel cells.
I would suggest the authors take a look at the following papers to improve the introduction of the current paper:

A. Bozorgnezhad, M. Shams, G. Ahmadi, H. Kanani, and M. Hasheminasab, “The Experimental Study of Water Accumulation in PEMFC Cathode Channel,” in ASME/JSME/KSME 2015 Joint Fluids Engineering Conference, Jul. 2015, pp. AJKFluids2015-22299, V001T22A004. doi: 10.1115/AJKFluids2015-22299.

H. Kanani, M. Shams, M. Hasheminasab, and A. Bozorgnezhad, “Model development and optimization of operating conditions to maximize PEMFC performance by response surface methodology,” Energy Convers. Manag., vol. 93, pp. 9–22, Mar. 2015, doi: 10.1016/J.ENCONMAN.2014.12.093.

M. Ashrafi, M. Shams, A. Bozorgnezhad, and G. Ahmadi, “Simulation and experimental validation of droplet dynamics in microchannels of PEM fuel cells,” Heat Mass Transf., vol. 52, no. 12, pp. 2671–2686, Dec. 2016, doi: 10.1007/s00231-016-1771-z.

A. Bozorgnezhad, M. Shams, H. Kanani, M. Hasheminasab, and G. Ahmadi, “The experimental study of water management in the cathode channel of single-serpentine transparent proton exchange membrane fuel cell by direct visualization,” Int. J. Hydrogen Energy, vol. 40, no. 6, pp. 2808–2832, Feb. 2015, doi: 10.1016/J.IJHYDENE.2014.12.083.

A. Bozorgnezhad, M. Shams, H. Kanani, M. Hasheminasab, and G. Ahmadi, “Two-phase flow and droplet behavior in microchannels of PEM fuel cell,” Int. J. Hydrogen Energy, vol. 41, no. 42, pp. 19164–19181, Nov. 2016, doi: 10.1016/J.IJHYDENE.2016.09.043.

M. Hasheminasab, A. Bozorgnezhad, M. Shams, G. Ahmadi, and H. Kanani, “Simultaneous Investigation of PEMFC Performance and Water Content at Different Flow Rates and Relative Humidities,” in ASME 2014 12th International Conference on Nanochannels, Microchannels and Minichannels, Aug. 2014, p. V001T07A002. doi: 10.1115/ICNMM2014-21586.

A. Bozorgnezhad, M. Shams, H. Kanani, and M. Hasheminasab, “Experimental Investigation on Dispersion of Water Droplets in the Single-Serpentine Channel of a PEM Fuel Cell,” J. Dispers. Sci. Technol., vol. 36, no. 8, pp. 1190–1197, Aug. 2015, doi: 10.1080/01932691.2014.974810.

Experimental design

The research is within the scope of the journal, specifically the Environmental Sciences, and the research question is well defined, relevant, and meaningful. The knowledge gap was identified and properly addressed as follows:
According to the submitted manuscript, in most research in the electrocoagulation process, the electrodes are vertical, while in the current research, they are installed horizontally to use the flotation power, and the optimization is by the single factorial method. This is a wise and clever idea, and the authors have well studied it. The experiment design was considered with three essential parameters: removal efficiency, electrode dissolution, and energy consumption; which completely make sense.
They did an excellent job with a robust and solid investigation performed at a high technical level using the ANN in their field. They indicated that machine learning-based models can be very effective at predicting and may even replace experimental and numerical techniques, which is outstanding research to contribute to the field of environmental sciences. The methods and information are mentioned in an excellent way.

Validity of the findings

The paper is really novel in environmental engineering and valid based on its conclusions and methods.

Additional comments

Overall, I highly recommend this paper to anyone interested in the optimization of wastewater treatment systems or the use of artificial neural networks in environmental engineering. The authors have made a significant contribution to the field, and their work has the potential to lead to improvements in wastewater treatment processes in the oil industry and beyond.

---

## Round 0.2 · Minor Revisions

The abstract section should include the motivation, methods, and major findings of the paper. Moreover, the literature review section should be enhanced and updated. The introduction section should include a mini review to discuss the literature gap and the contribution of the research. The language of the paper also needs attention.

---

## Round 0.3 · accepted · Accept

The paper is improved after all revisions. Particularly, my comments on the last version are fully addressed. Thus, the paper is accepted for publication.